

# Forest aboveground biomass stock and resilience in a tropical landscape of Thailand

Nidhi Jha[1,2], Nitin Kumar Tripathi[1], Wirong Chanthorn[3], Warren Brockelman[4], Anuttara Nathalang [4,5], Raphaël Pélissier[2], Siriruk Pimmasarn[1], Pierre Ploton[2], Nophea Sasaki[6], Salvatore G.P. Virdis[1], Maxime Réjou-Méchain[2]

[1]Field of Remote Sensing and GIS (RSGIS), Department of Information & Communication Technologies, Asian Institute of Technology, Thailand.

[2]AMAP IRD, CNRS, CIRAD, INRA, Univ Montpellier, Montpellier, France.

[3]Faculty of Environment, Kasetsart University, Thailand

[4]National Center for Genetic Engineering and Biotechnology (BIOTEC), Thailand

[5]National Biobank of Thailand (NBT), Pathum Thani, Thailand

[6]Department of Development and Sustainability, Asian Institute of Technology, Thailand

*Correspondence to*: Nidhi Jha (nidhi23aug@gmail.com)

**Abstract.** Half of Asian tropical forests were disturbed in the last century resulting in the dominance of secondary forests in Southeast Asia. However, the rate at which biomass accumulates during the recovery process in these forests is poorly understood. We studied a forest landscape located in Khao Yai National Park (Thailand) that experienced strong disturbances in the last century due to clearance by swidden farmers. Combining recent field and airborne laser scanning (ALS) data, we first built a high-resolution aboveground biomass (AGB) map over 60 km$^2$ of the forest landscape. We then used the random forest algorithm and Landsat time-series (LTS) data to classify landscape patches as non-forested versus forested on an almost annual basis from 1972 to 2017. The resulting chronosequence was then used in combination with the AGB map to estimate forest carbon recovery rates in secondary forest patches during the first 42 years of succession. The ALS-AGB model predicted AGB with an error of 14% at 0.5-ha resolution (RMSE = 45 Mg ha$^{-1}$) using the mean top-of-canopy height as a single predictor. The mean AGB over the landscape was of 291 Mg ha$^{-1}$ showing a high level of carbon storage despite past disturbance history. We found that AGB recovery varies non-linearly in the first 42 years of the succession, with an increasing rate of accumulation through time. We predicted a mean AGB recovery rate of 6.9 Mg ha$^{-1}$ yr$^{-1}$, with a mean AGB gain of 143 and 273 Mg ha$^{-1}$ after 20 and 40 years, respectively. These estimates are within the range of those reported for the well-studied Latin American secondary forests under similar climatic conditions. This study illustrates the potential of ALS data not only for scaling up field AGB measurements but also for predicting AGB recovery dynamics when combined with long-term satellite data. It also



illustrates that tropical forest landscapes that were disturbed in the past are of utmost importance for the regional carbon budget and thus for implementing international programs such as REDD+.

## 1 Introduction

Tropical forest disturbances and subsequent biomass recovery through time significantly affect the global carbon cycle (Harris et al., 2012). Although secondary forests in the tropics could constitute a major global carbon sink, the magnitude of such sink remains poorly understood (Chazdon, 2014; Lugo and Brown, 1992). The previous study has found that this carbon sink is accountable for absorbing half of the carbon emissions from fossil fuels and industrial production at the scale of Latin America (Chazdon et al., 2016). Thus, there has been much interest in quantifying the ability of tropical secondary forests to sequester carbon in order to reduce uncertainties in the global carbon balance (e.g., Chai, 1997; Lohbeck, Poorter, Martínez-Ramos, & Bongers, 2015; Stas, et al., 2017).

Previous studies have used long-term forest plot surveys along chronosequences to quantify forest carbon dynamics in secondary tropical forests (Chazdon et al., 2007; N'Guessan et al., 2019; Norden et al., 2011, 2015; Poorter et al., 2016a; Rozendaal and Chazdon, 2015). Although long-term forest plots are essential for understanding the dynamics of tropical forests (Losos and Leigh, 2004), they are scarce, inherently labor-intensive, expensive and time-consuming to maintain, and not evenly distributed in the tropics. In addition, most studies of carbon dynamics along tropical forest successions are concentrated in Latin America (Chave et al., in revision; Letcher and Chazdon, 2009; Norden et al., 2015; Poorter et al., 2016a; Rozendaal et al., 2017; Rozendaal and Chazdon, 2015 but see N'Guessan et al., 2019 in Africa). They show high among-site variation in forest carbon recovery rates, suggesting a high context-dependence (Chazdon et al., 2007; Norden et al., 2011, 2015), partly depending on climate conditions (Poorter et al., 2016a). However, pantropical studies have shown that the carbon potential of Latin American forests is smaller than that of Southeast Asian and African forests (Feldpausch et al., 2012; Sullivan et al., 2017). Whether Latin American estimates of forest carbon recovery dynamics can be generalized to forests at the pantropical scale is thus highly uncertain. This issue is especially crucial for Asian tropical forests where half of the forests have been disturbed during the last century, resulting in the dominance of secondary forests throughout the region (Achard et al., 2014; Mitchard et al., 2013; Stibig et al., 2014).

Remote sensing technology has emerged as a promising tool for extrapolating local field carbon estimates over landscapes, regions, or at the global scale (Gibbs et al., 2007; Goetz et al., 2009). However, current long-term (>20 years) satellite data such as Landsat are weakly sensitive to forest carbon, especially in high-biomass forests (Ferraz et al., 2018; Lu, 2006; Meyer et al., 2019; Zheng et al., 2004). Yet, these data may be used to produce reliable land-cover classifications (e.g., forest versus non-forest areas; FAO 2010). They allow to assess the dynamics of deforestation and reforestation worldwide (Hansen et al., 2013) and can thus monitor disturbance history, particularly the time since abandonment of agriculture (Cohen et al., 1996; Masek and Collatz, 2006). However, the forest carbon dynamics associated with such deforestation and





reforestation events remains highly uncertain due to the large uncertainties of global carbon maps (Mitchard et al., 2013, 2014; Réjou-Méchain et al., 2019).

On another hand, airborne laser scanning (ALS) provides accurate landscape-scale estimates of forest structural
parameters (Maltamo et al., 2005; Næsset, 2002; Wulder et al., 2012). When calibrated with field-based estimates of aboveground biomass (AGB), ALS metrics can be used to produce high-resolution forest carbon maps, even for high carbon-dense tropical forests (Asner et al., 2010; Cao et al., 2016; Ferraz et al., 2018; Kronseder et al., 2012; Labriere et al., 2018; Zhao et al., 2009; Zolkos et al., 2013). Multi-temporal ALS acquisitions may thus provide direct estimates of the carbon balance of tropical forest landscapes (Dubayah et al., 2010; Meyer et al., 2013; Réjou-Méchain et al., 2015). However, due to
its relatively recent emergence, ALS technology cannot be used to investigate long-term dynamics directly yet (>10 years).

Combining long-term (>40 years) land cover change assessment from satellite data archives (e.g., Landsat) and contemporary LiDAR AGB maps may be a promising avenue for understanding the long-term forest carbon dynamics. Such an approach has been successfully implemented in temperate and boreal forests (Bolton et al., 2015; Pflugmacher et al., 2012, 2014; White et al., 2018; Zald et al., 2014). However, to our knowledge, it has not been yet used to assess the forest carbon
resilience of tropical forests (but see Helmer et al., 2009 who used satellite-based LiDAR).

In this study, we combined extensive field, ALS, and LTS data to assess the spatial variation of AGB and forest AGB dynamics of secondary forests in a Thai landscape. More specifically, we first calibrated a robust ALS-AGB model to produce a fine-scale AGB map at the landscape scale. We then used a random forest machine-learning algorithm to classify historical Landsat images from 1972 to 2017 into forest and non-forest classes. Using this information over time, we generated a
cumulative forest gain map over a period of 42 years of succession. We finally combined this chronosequence with our ALS-AGB map to estimate the forest carbon resilience of secondary forests during the 42 first years after land abandonment.

## 2. Materials and methods

### 2.1 Study area

The study area of ca. 6,400 ha is part of Khao Yai National Park in central Thailand (latitude: 14° 25' 20.4" N, longitude: 101°
22' 36.9" E; Fig. 1). Khao Yai is the first national park of Thailand, established in 1962. It is home to numerous endangered plant and animal species (Kitamura et al., 2004). The area receives approximately 2,200 mm of precipitation annually, with a dry season of five to six months (precipitation below 100 mm month$^{-1}$) from November to April (Brockelman et al., 2011; Chanthorn et al., 2016). The annual mean temperature is about 22–23°C (Jenks et al. 2011), and the altitude of the study area varies from 650 m to 870 m. Before establishment of the park, some areas were used for low-intensity agriculture activities
(Brockelman et al., 2011, 2017) and then naturally reforested at different times depending on when burning ceased (Chanthorn et al., 2016). As a consequence, the landscape constitutes a mosaic of secondary forests of different ages amidst old-growth forests (Chanthorn et al., 2016).





## 2.2 Field data

We used three sets of forest inventory plots with a total sample area of 35 ha (Fig. 1). First, a large 30-ha contiguous (500 m × 600 m) forest dynamics plot, named Mo Singto, was established in old-growth forest after 1998 and completely censused in 2004–2005, 2010–2011 and 2016–2017. The census method follows the protocol of the Center for Tropical Forest Science (CTFS) network which the plot officially joined in 2009 (Brockelman et al., 2011). The second set of plots included eight separate 0.48-ha plots (60 m × 80 m) that were established from March to May 2013 and re-censused from November 2017 to

January 2018 (Chanthorn et al., 2017). These plots are set along a successional gradient varying from near stand initiation to old-growth forest. Lastly, a 1-ha plot (100 m × 100 m) located near the north border of the 30-ha Mo Singto plot was established in a secondary forest in 2005 and then re-censused in 2010 and 2017. In all plots, trees ≥1 cm in diameter at breast height (dbh) were tagged, identified to species, mapped and measured for their diameter, except in the 0.48-ha plots where the minimum dbh was 4 cm. A total of 184,239 individual trees were measured across all the plots, from which 517 trees were measured for

height using a pole for short trees (<5 m), a laser range finder (Nikon Forestry 550) for medium height trees (5–7m) and a Vertex III hypsometer for tall (>7 m) trees (Chanthorn et al., 2017). In this paper, we used the 2017 census data, concomitant with the ALS campaign, to estimate AGB and multiple censuses to estimate the AGB dynamics of secondary plots. For the sake of homogeneity in tree measurements, we used only trees ≥5 cm in dbh in the whole dataset.

In order to homogenize plot size, we subdivided all plots ≥1 ha into 0.5-ha subplots. This resulted in 70 plots of either

50 m × 100 m (n = 62) or 60 m × 80 m (n = 8) that we classified in three successional stages from young to old-growth forests following the classification from (Chanthorn et al., 2017): Stand initiation (early) stage (SIS; n = 3); stem exclusion (intermediate) stage (SES; n = 5), and old growth stage (OGS; n = 62). Based on interviews of senior park rangers and using Landsat remote sensing data, (Chanthorn et al., 2017) estimated that the age of the forests was approximately 15–20 years for SIS forests, 35–40 years for SES forests and unknown but probably older than 200 years for OGS forests.

## 2.3 ALS data

The airborne laser scanning (ALS) campaign was conducted on 10 April 2017 over ca. 64 km² (Fig. 1). The Asian Aerospace Services Limited company (Bangkok) acquired the ALS data with a RIEGL LMS Q680i installed into a Diamond Aircraft "Airborne Sensors" DA-42 fixed-wing plane. The flying altitude was about 500–600 m above ground level with a 60-degree field of view, and a pulse repetition frequency of 400 kHz, for which the aircraft maintained an average ground speed of 185

km hr$^{-1}$ capturing the area of interest in 50 overlapping laser strips. We discretized the full waveform data for subsequent analyses resulting in an average point density of ca. 22 points m$^{-2}$.

Post-processing of ALS data and point cloud classification into the ground, vegetation, or noise were done using TerraScan of Terrasolid Version 14. Points classified as ground were used to build a digital terrain model (DTM) at 1-m resolution using a k-nearest neighbour kriging approach implemented in the LidR R package (Roussel and Auty, 2017). A 1-

m resolution canopy height model (CHM) was then computed from the height of the normalized vegetation points, discarding



outliers classified as air or noise. Finally, we used the CHM and the normalized vegetation point cloud to derive different forest height metrics (L-metrics) at the plot level (Table S1).

## 2.4 Landsat data

We retrieved Landsat images (MSS, TM, OLI and TIRS products) for the study area from the Landsat archive
(**http://glovis.usgs.gov**) between the 1972–2017 period (WRS-1 138/50 and WRS-2 path/row: 129/50). To minimize the impact of clouds and potentially varying phenology within years, we mostly selected images acquired during the dry season, from November to March. We thus collected Landsat 1-3 MSS data (1972–1983), Landsat 4-5 TM (1984–2011), and Landsat 8 OLI & TIRS (2013–2017) data (Supplementary Info, S3). We did not consider Landsat 7 ETM+ images due to the failure of the Scan Line Corrector, leading to data gaps. All Landsat images were already orthorectified and displayed an accurate co-
registration with ALS data. Before 1984, Landsat MSS collected data at 60 m × 60 m spatial resolution in most bands. Thus, to have consistent time series data, we aligned all the post-1983 Landsat data using a reference image from 1972 and aggregated each image to 60 × 60 m. Over the 44 years, we selected a total of 34 high-quality images, each representing one year. For the 11 missing years, we cannot find cloud-free images and no image was available in 2012 since we discarded Landsat 7 ETM+ data.

**2.5 Field aboveground biomass calculation**

We estimated tree aboveground biomass (AGB) using a pantropical allometric model (Eq. 4 from Chave et al., 2014). This model uses the diameter (D), total tree height (H) and wood density (WD) as the predictors and was shown to hold across tropical vegetation types and regions. Wood density was estimated using species (47% of stems), genus (50%) or stand (3%) averaged values from the global wood density database (Chave et al., 2009; Zanne et al., 2009). Tree height was estimated
through locally-adjusted height-diameter (H-D) models of the form given in Eq. (1):

$$\ln(H) = a + b \times \ln(D) + c \times \ln(D)^2 + \varepsilon \qquad (1)$$

where $a$ and $b$ are model parameters to be adjusted and $\varepsilon$ is a normally distributed error with mean 0 and standard error $\sigma logH$. Tree height was subsequently estimated using the back-transformation formula including a known bias correction (Baskerville, 1972) using following Eq. (2):

$$H = \exp(0.5 \times \sigma_{logH}^2 + a + b \times \ln(D) + c \times \ln(D)^2 + \varepsilon) \qquad (2)$$

Because H-D relationship varies along the successional gradient (Chanthorn et al., 2017), we fitted three independent H-D models for the three different successional growth forest stages using 177 measured trees for SIS plots, 159 for SES plots and 181 for OGS plots.

AGB at the plot level was then estimated in Mg ha⁻¹ by summing individual tree AGB for all trees belonging to the
plot. We did all these analyses using the R BIOMASS package (Réjou-Méchain et al., 2017).





## 2.6 LiDAR AGB model

We relied on a log-log model form given in Eq. (3) to model AGB from ALS data (Asner et al., 2012; Réjou-Méchain et al., 2015):

$$Ln(AGB) = a + b \times \ln(L1) + c \times \ln(L2) + \dots + \varepsilon \qquad (3)$$

Where $L1$, $L2$, … are the LiDAR metrics to be selected (see below) and $\varepsilon$ the error term assumed to be normally distributed with zero mean and residual standard error $\sigma logL$. Fitting the model with log-transformed variables allows us to model a multiplicative error and thus to account for higher model prediction error with larger AGB values (Zolkos et al., 2013). Using this model, we selected the most predictive LiDAR metrics from our full set of LiDAR metrics using a leave-one-out-cross-validation (LOOCV) scheme nested within a forward selection procedure. The LOOCV consists of fitting models with all

observations except one, and then using the model to predict the value of the observation held out of model calibration. The process is repeated for all observations so that model prediction accuracy, here the root mean squared error (RMSE), can be independently assessed from all observations. This LOOCV approach was repeated for different models following a forward procedure that begins by selecting the most discriminant variable according to the LOOCV-RMSE criterion. The procedure then continues by selecting the second most discriminant variable and so on. To minimize the problem of model overfitting,

we only kept explanatory variables that contribute to a decrease in relative RMSE (RMSE divided by the mean observed AGB) by more than 1%. The selected LiDAR-AGB model was then used to predict AGB values over the Landscape at 60-m resolution, to match the resolution of Landsat images.

## 2.7 AGB recovery analysis

### 2.7.1 Forest-non-forest classification

To classify areas as forest or non-forest, we applied the random forest (RF) algorithm independently on each Landsat image to minimize inter-images classification error that may otherwise arise from instrumental (e.g. differences in sensors spectral characteristics) and phenological effects. We used all Landsat bands and their ratios as predictors in our RF classification models i.e. the 4 raw bands for Landsat 1-3 MSS data (1972-1983), the 7 raw bands for Landsat 4-5 TM (1984-2011) and the 9 raw bands for Landsat 8 OLI & TIRS (2013-2017). The normalized difference vegetation index (NDVI) was additionally

used as a predictor for all the sensors while the normalized burn ratio (NBR) was only used for Landsat 4-5 and Landsat 8 due to non-availability of SWIR bands in MSS sensors. Thus, we used 18 predictors for MSS, 51 predictors for TM and 83 for OLI & TIRS as an input for the RF algorithm. RF model for each year of the study period was then trained on the same set of pixels that likely remained either forested or non-forested from 1972 up until 2017. This training dataset was built using the 2017 ALS data. We first aggregated the 1-m LiDAR-derived CHM at the same resolution as the Landsat images (60-m

resolution) and defined non-forest pixels as pixels having a mean top of canopy height < 5 m (FAO, 2012; Sasaki & Putz, 2009). Because 60-m scale deforestation is unlikely to have occurred in the area since the establishment of the national park in 1962, areas that were classified as non-forest with the 2017 LiDAR data very likely corresponded to non-forested areas





during the whole study period. By contrast, we defined as forested areas all 60-m pixels that had a LiDAR mean top of canopy height > 30 m because these tall forests very likely corresponded to forested areas during the whole study period. We thus used

a reference set of 400 60-m pixels classified as non-forest and 110 as forest. This dataset was then randomly divided into a training dataset (60%) to calibrate the RF models and a validation dataset (40%) to assess the accuracy of the forest and non-forest classification. We only considered classified pixels that had a post-probability of assignment >90% in the RF outputs (Pickell et al., 2016; White et al., 2018) and calculated the classification accuracy as the proportion of pixels that were correctly classified as forest or non-forest in the validation dataset. This statistical analysis was done using the "randomForest" R

package (Liaw and Wiener, 2002).

### 2.7.2 Forest AGB recovery analysis

We combined time-series classified Landsat images with the 60-m resolution LiDAR AGB map to quantify AGB recovery as a function of time. We used classified time-series data to assign to each pixel the last date at which a shift from a non-forest to forest status occurred during the study period. Thus, all pixels that did not experience any shift, i.e. that remained non-forested

or forested during the whole study period were discarded from this analysis. To minimize false detection of land cover change due, for example, to atmospheric pollution, we only considered shifts that entailed land cover change for at least two consecutive images. Thus, we did not consider any shift before 1975 because, to be considered, the non-forest to forest shift of a pixel should occur after being classified as non-forest in the two previous images (in our case in 1972 and 1973). Finally, we also discarded pixels that underwent more than four different shifts during the whole study period because numerous shifts

are likely to indicate areas prone to forest degradation, e.g. close to human occupancy areas such as roads, introducing a bias in our inferences on the natural successional dynamics. We thus were able to assign to the selected pixels their last forest recovery starting time and the corresponding AGB predicted by the LiDAR AGB map to quantify AGB recovery as a function of time at 60-m resolution.

### 3. Results

#### 3.1 Forest biomass stocks

Field plots AGB ranged from 80 to 577 Mg ha$^{-1}$ (mean of 315 Mg ha$^{-1}$), with a mean AGB of OGS, SES and SIS plots of 328 Mg ha$^{-1}$, 291 Mg ha$^{-1}$, and 87 Mg ha$^{-1}$, respectively. Among all the LiDAR metrics, the mean of top-of-canopy height (TCH, defined as the maximum height of 1-m resolution pixels) was the best predictor of field AGB estimates with a relative RMSE of 14% (RMSE = 45 Mg ha$^{-1}$) at 0.5-ha scale (Fig. 2). Adding a second predictor did not reduce the relative RMSE by more

than 1% (Table S2). We thus kept TCH as a single predictor for our analyses resulting in the following Eq. (4) for LiDAR-AGB model:

$$AGB_L = 4.30 \times TCH^{1.39} \qquad\qquad (4)$$





Using this LiDAR-AGB model, we predicted AGB over the whole landscape (Fig. 3a). The distribution of AGB values over the landscape was not normally distributed due to an over-representation of pixels with low AGB values. At the
landscape scale, predicted AGB ranged from 0 to 681 Mg ha$^{-1}$ with a mean of 291 Mg ha$^{-1}$ (Fig. 3b), close to the mean AGB of the field plots.

**3.2 AGB recovery analysis**

Our forest and non-forest classification through time was highly accurate, with 90% to 99% of well-classified validation pixels in individual classified images (Table S3 & Fig. S1). Figure 4a illustrates the resulting spatialized time series of non-forest-to-
forest shifts over the study area and showed that most (83%) of the landscape did not experience such shift at 60-m resolution over the 42-year study period. Over the 17% remaining pixels that experienced a shift, we concentrated our analyses on the 4% pixels (n = 550; ca. 198 ha) that passed our selection procedure and that were well distributed over the landscape (Fig. 4a). Of all these selected pixels almost 60% of the shifts occurred before 1990 (Fig. 4b).

Considering the selected pixels that experienced a shift from non-forest to forest, we found that AGB accumulated
non-linearly through time during the 42 first years of the succession (Fig. 5). A simple power model led to a pseudo-R$^2$ of 0.66 and a power exponent greater than 1, indicating an increase in the rate of AGB accumulation with recovery time. This model predicts an AGB gain of 143 Mg ha$^{-1}$ after 20 years of recovery and of 273 Mg ha$^{-1}$ after 40 years (spatialized gain in AGB is shown in Fig. S2). Finally, using field AGB estimates from eight secondary forest plots at two different census dates, we showed that the observed rate of AGB accumulation was similar to the one predicted by our model and also tended to increase
with forest age (in blue dots in Fig. 5).

**4. Discussion**

In this study, we showed that the integration of field inventory, Landsat archives, and LiDAR data provide a powerful approach for characterizing the spatio-temporal dynamics of aboveground biomass in tropical forests. While the carbon stocks and recovery potential of south-Asian tropical forests are globally poorly known, our approach contributes to a better understanding
of the role of these forests in global carbon dynamics. We specifically showed that our study site stores a large amount of carbon, despite its disturbance history, and probably acts as a strong carbon sink, through secondary succession pathways.

**4.1 Spatial variation in AGB**

Using extensive field data, we have shown that forest AGB can be predicted with an error of 14% at a 0.5-ha resolution using a single LiDAR metric, the mean top-of-canopy-height (TCH), a metric previously identified as a robust predictor of AGB
(Asner and Mascaro, 2014). This error typically falls within the range of expected errors at this resolution (Zolkos et al., 2013). Using a robust metric selection approach, we showed that adding any other LiDAR metrics did not bring any additional information and that our single predictor did not show any saturation for large AGB values. Many studies have used a





combination of several LiDAR metrics selected through less robust approaches, i.e. not through independent validation approaches such as our LOOCV procedure, potentially generating overfitting problems (Junttila et al., 2015). We here confirm, similarly to Asner et al. (2012) and Réjou-Méchain et al. (2015), that simple parsimonious models should be preferred, at least within a given tropical forest landscape.


Using this LiDAR model, we predicted a mean AGB over the landscape of 291 Mg ha⁻¹, corresponding to a carbon density of 137 MgC ha⁻¹ (using a ratio of biomass to carbon conversion of 0.47; Thomas and Martin, 2012). Using a large network of field plots, a recent pantropical study suggested that Southeast Asian and African forests store significantly more carbon than Amazonian forests (Sullivan et al., 2017). However, in this latter study, Southeast Asian forests were only represented by field data from Indonesia and Malaysia where trees are known to be particularly tall (Coomes et al., 2017; Feldpausch et al., 2011; Jucker et al., 2017, 2018). Here, we found that our study forests stored significantly less carbon than forests in Indonesia and Malaysia, where the mean carbon density reached ca. 200 MgC ha⁻¹ (Sullivan et al., 2017), but as much as in Amazonian forests (mean of 140 MgC ha⁻¹; Sullivan et al., 2017), even when considering only old-growth forest plots. Whether the relatively low carbon density of our study site, compared to other Southeast Asian forests, is specific to our study area or representative of other Southeast Asian forests should be further investigated.



We found a very high spatial heterogeneity of AGB at the landscape scale with an apparent over-representation of low AGB values. This is most probably the consequence of past human activities in this area up to the establishment of the park that led to the present mosaic of secondary and mature forests. This result indicates that this area is currently likely to be a net carbon sink.


## 4.2 AGB recovery through time

Combining classified images obtained from LTS and LiDAR data, we quantified the recovery rate of forests after land abandonment. As expected, we showed a significant increase of AGB with recovery time. After 20 years of recovery, our model predicts an AGB accumulation of 143 Mg ha⁻¹, an estimate slightly higher than the one predicted by Poorter et al., (2016a) in Neotropical secondary forests (122 Mg ha⁻¹). This corresponds to a net carbon uptake of 3.4 Mg C ha⁻¹yr⁻¹ for the first 20 years and of 3.2 Mg C ha⁻¹yr⁻¹ for the first 40 years. This rate of carbon accumulation over the first 20 years of succession is close to the pantropical estimate from Silver et al., (2000).


Our model showed that a non-linear power model with an exponent > 1 best fit our data, suggesting an increase in the rate of carbon accumulation during the first 42 years of succession. Contrary to the results found by Feldpausch et al. (2007), the rates of AGB accumulation inferred with our approach provided estimates similar to those obtained from long-term field plot surveys (Fig. 5), validating the chronosequence approach in our study area. Assuming that the carbon recovery rate rapidly decreases after 50–60 years (Brown and Lugo, 1990; Silver et al., 2000), our result suggests a sigmoid relationship of AGB accumulation with time in our study area. Previous studies have shown different models of AGB accumulation with forest age. Saldarriaga et al. (1988) showed that the AGB of Neotropical forests from the upper Rio Negro increased linearly with stand age during the 40 years, while Jepsen, (2006) reported a sigmoidal increase in AGB accumulation in Sarawak, Malaysia, as is







likely the case in our study area. Finally, working on 41 Neotropical sites, Poorter et al. (2016a) assumed a logarithmic trend in the AGB accumulation over time, hence a decrease of the rate of carbon accumulation through time, probably because they investigated a longer time period. Selecting the sites of Poorter et al. (2016a, 2016b) that had at least 10 observations over the first 44 years (n = 21 out of 28 sites, i.e. excluding 7 sites for which model fitting was not possible), site-specific power models revealed that two-thirds of the sites displayed a power exponent <1 and one-third showed an exponent >1 (Fig. S3). Thus, the accumulation of AGB with age follows different trends across sites, as already highlighted in previous studies (Kennard et al., 2002; Poorter et al., 2016a; Ray and Brown, 2006; Ruiz et al., 2005; Silver et al., 2000; Toledo and Salick, 2006). Understanding how these trends vary according to abiotic factors (e.g. soil type, rainfall), species assemblage and diversity, or to priority effects such as types of land use and land management existing before forest recolonization, constitutes an important research perspective (Chazdon, 2014; McMahon et al., 2019).

Our analysis was based on a forest/non-forest classification through time and our independent validation suggested a high overall accuracy (90 to 99%), similar to that reported by other studies using Landsat data classification in boreal systems (Bolton et al., 2015; White et al., 2018). Furthermore, our estimate of forest age using this approach was highly consistent with our expectations. Indeed, using our forest plots, we found that the SES and SIS forest stages lasted on average 40 years (range 38–42) and 13 years (range 8–20), respectively, hence very close to suggestions of Chanthorn et al. (2017) (Fig. S4). However, our overall approach cannot be replicated easily in human-occupied areas. Indeed, human disturbances lead to forest degradation that, in contrast to deforestation, is not captured by the Landsat signal, so that, when combined with a reference AGB map, natural carbon recovery potential could be seriously underestimated. Because our study area was protected from human disturbances during the study period, we were in very favourable conditions to estimate forest carbon recovery rates and strongly encourage researchers benefiting from similar conditions to replicate our analyses in other study sites.

## 5 Conclusions

Our study demonstrates that combining field, LiDAR, and long-term satellite data provides an efficient way to assess forest carbon recovery rates during secondary successions. We showed that it produces similar estimates as those inferred from long term field plots, but at a much lower cost and within a much shorter time frame. Replicating this approach in other protected tropical landscapes would thus considerably increase the representativeness of forest carbon recovery rates and would probably improve our understanding of the environmental and historical drivers of these varying rates. This is especially important in Southeast Asian forests that constitute a hotspot of biodiversity and carbon, and that are under threat due to the fast changing of both environment and socio-economics in this region. Quantifying the rates at which different forest types accumulate carbon should thus stay at the forefront of the research agenda and would greatly benefit the Earth system model community and international policy initiatives such as REDD+.



**Code availability/Data availability**

Code and data are available upon request to the corresponding author.

**Author contribution**

NJ, NKT, and MRM designed the study; NJ and NRM analyzed the data and wrote the first draft of the paper; WC, WB, and AN provided field data. All authors provided valuable feedbacks on analyses and the manuscript.


**Competing interests**

The authors declare no competing interest.

**Acknowledgments**

This study was supported by the project (AIT/SET - 2016 - R011) sponsored by the French Ministry of Foreign Affairs, and International Development initiated during the Regional Forum on Climate Change. We gratefully thank the National Science and Technology Development Agency (Thailand) for supporting the long-term monitoring of all forest plots and the Department of National Parks, Wildlife and Plant Conservation (DNP) that supported our research.

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



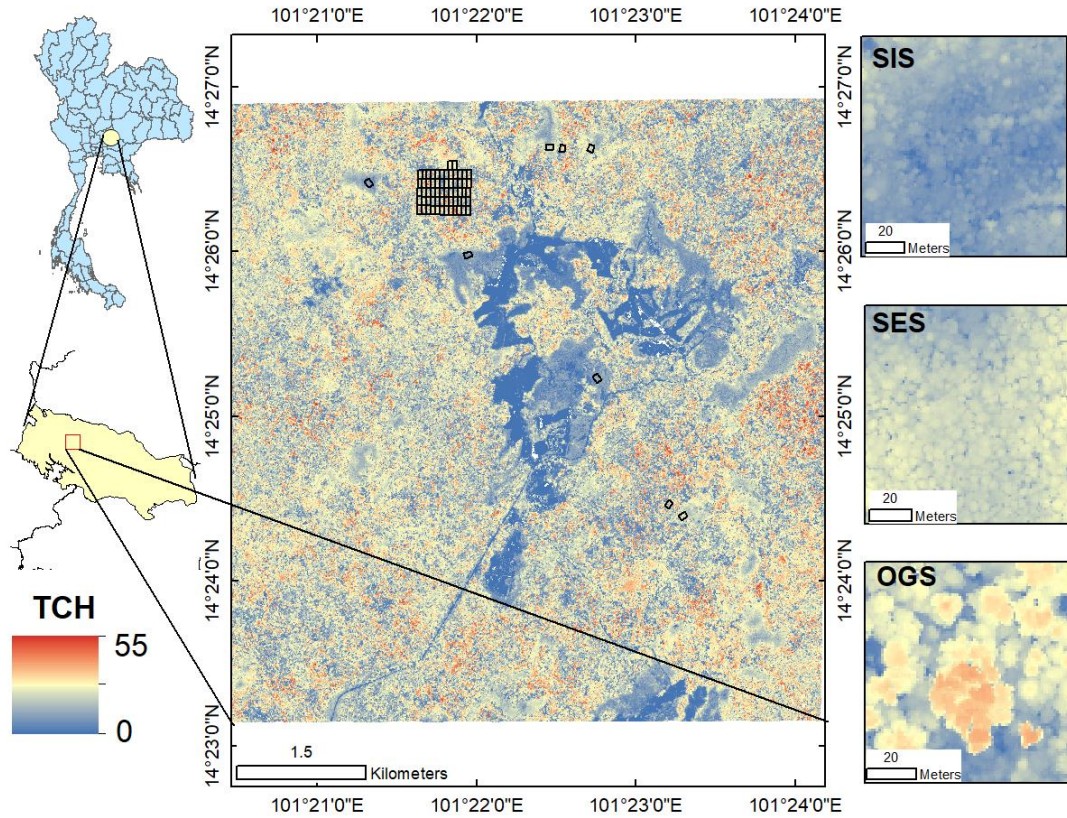

**Figure 1. Study area. Location of the study area in Thailand (upper left) and in the Khao Yai reserve (bottom left). The central map illustrates the LiDAR top of canopy height in the study area at 1-m resolution and the location of the 70 studied plots (in black). Examples of the different stand development stages are illustrated (right; SIS: stand initiation stage; SES: stem exclusion stage; and OGS: old growth stage).**






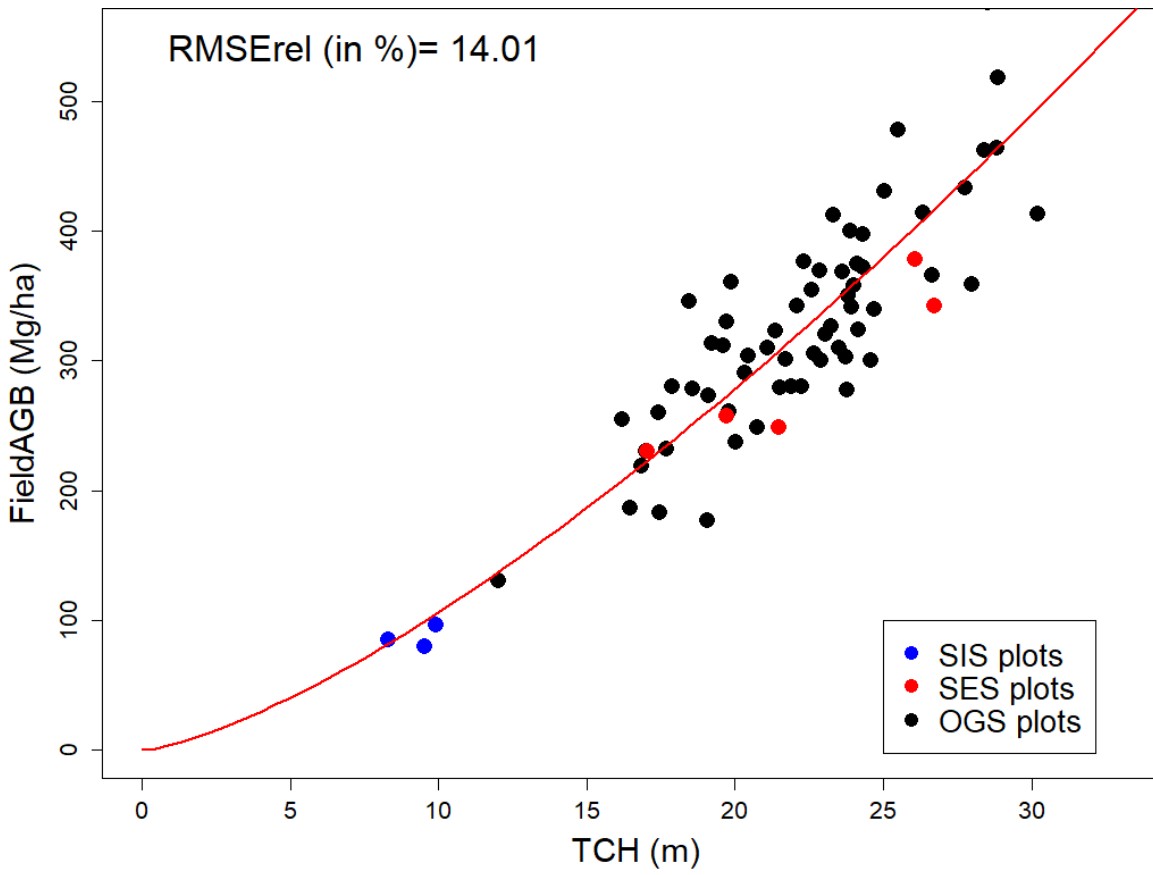

**Figure 2. LiDAR-AGB model showing the relationship between field-derived plot AGB and the LiDAR top-of-canopy height (TCH) at a 0.5-ha resolution. The power model is illustrated by the red line, and the successional types to which field plots pertain according to Wirong et al. (2017) are given in the legend.**





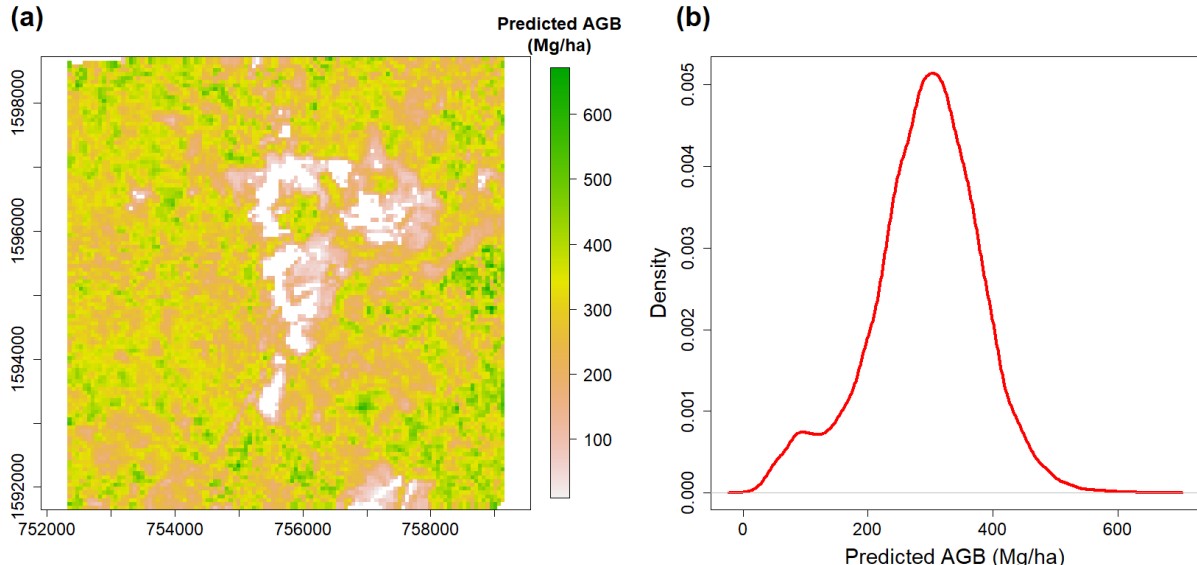

**Figure 3. LiDAR-AGB map and the distribution of AGB values over the landscape at 60-m resolution. (a)- Spatial distribution of AGB predicted from the LiDAR-AGB inversion model over the study area; (b)- Density distribution of predicted AGB over the landscape.**



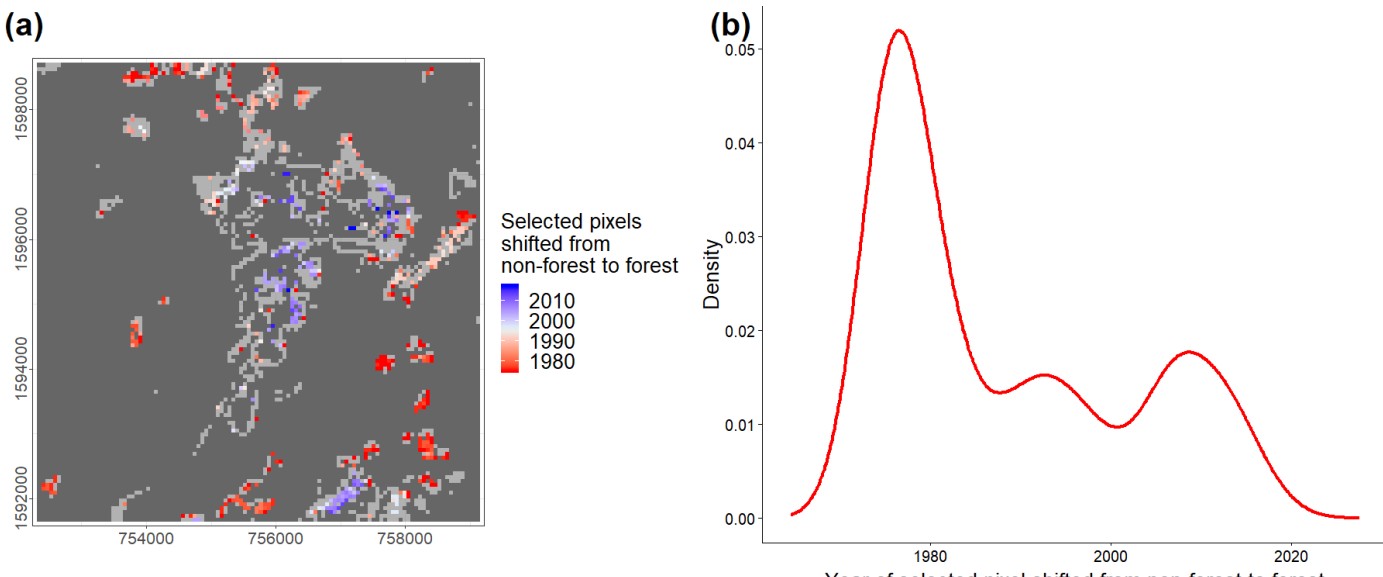


**Figure 4. Landsat time series derived map showing non-forest-to-forest change over the study area. (a)- Map showing spatialized selected pixel shifts from non-forests to forests over years. Dark grey represents pixels that did not experience any shift and light grey represents pixels which did not pass our quality procedure during the study period (b)- Density distribution of selected pixel shifts over the landscape during the study period.**





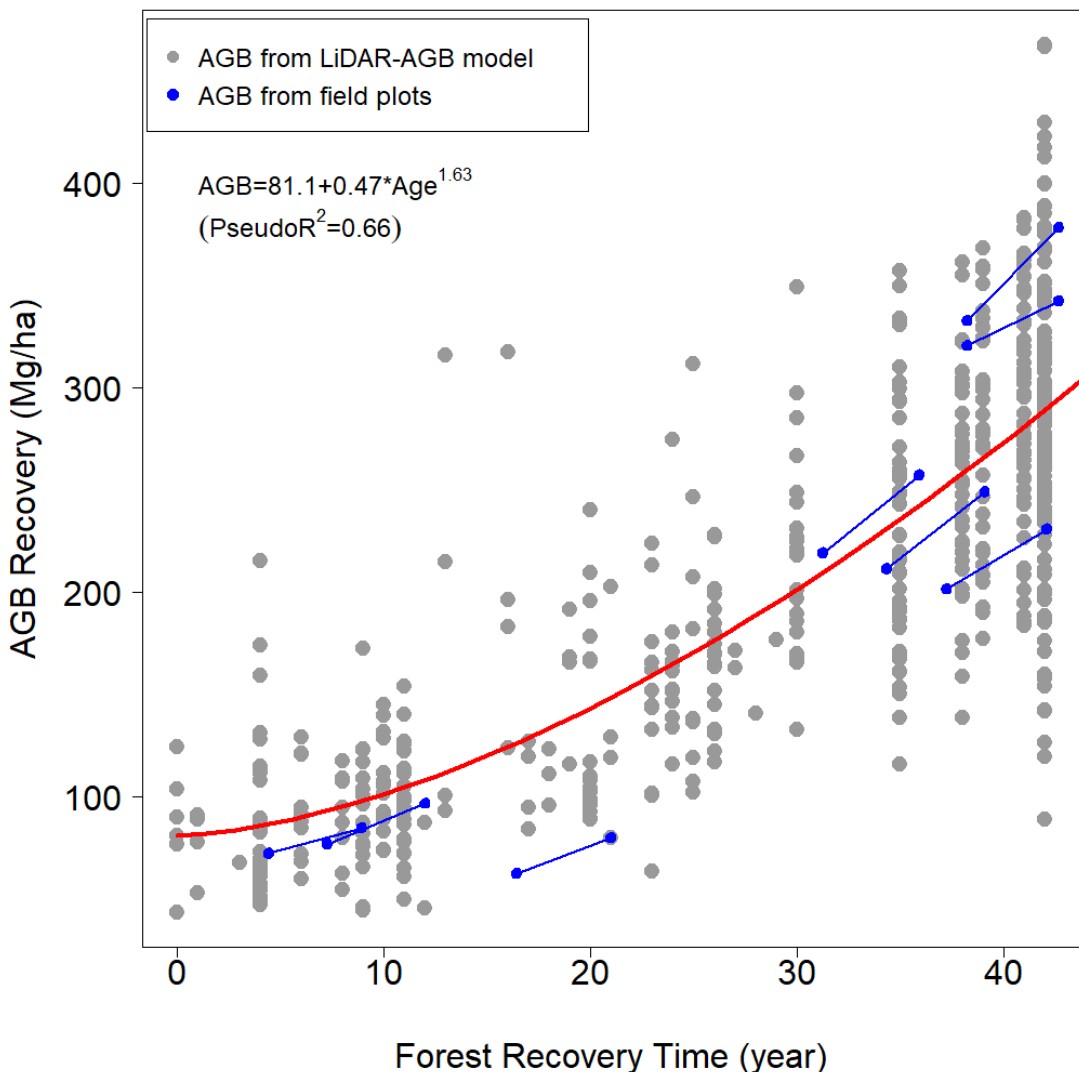


**Figure 5. Relationship between forest biomass estimated from a LiDAR-AGB model and forest recovery time estimated from a time series of classified Landsat images (grey dots). The fitted power model is represented by the red line. Blue lines and dots represent the AGB directly estimated from eight field plots (same plots are joined by a line) in 2013 and in 2017/8 and for which the forest recovery time was inferred from Landsat derived forest age (Fig. S4).**
