# Peer review of "Forest aboveground biomass stock and resilience in a tropical landscape of Thailand"

_Biogeosciences, 2019_

## Referee Comment (RC1) · Anonymous Referee #1 · 29 Aug 2019

This is a robust and well-written study that combines field measurements, multi-temporal satellite imagery, and airborne laser scanning data at the landscape scale to estimate rates of biomass accumulation in naturally regenerating forest vegetation in Khao Yai National Park in central Thailand. As such high-quality information is lacking from most regions of Asia, this study will be a landmark case and illustrates how combinations of different data sources can be used to track changes in landscape scale biomass accumulation and carbon storage in the absence of long-term monitoring data from forest sites. I applaud the authors on a job well done. One short-coming of their model is that very few field sites had low ABG values, so the model may not be as accurate at predicting AGB at low levels.

With the data that they have, the authors estimated the distribution of AGB values

across the landscape. These data were used to estimate mean landscape-scale AGB (and carbon density) for 2017. With the information on changing states of pixels from non-forest to forest (or from forest to non-forest), it should be possible to estimate how the distribution, mean, and total AGB within the landscape changed from the mid 1970s to the present day. This would be fascinating to do (if not in this paper, then in another one).

Additional comments:

Line 36-38: this statement does not describe what Chazdon et al. 2016 concluded. They found that 40 yr of carbon storage in regenerating forests of lowland regions of Latin American tropics alone offset the past 19 years of carbon emissions from fossil fuel burning and industrial sources from all of Latin America (not total carbon emissions).

Line 102: what is the age and prior land use of this secondary forest?

Line 112: were there any stands in the understory initiation phase? Some details from Chanthorn et al. 2017 should be included here.

Line 155: but only those > 5 cm dbh, right?

Line 241: I would take out the word "probably" Why wouldn't it? How has the carbon storage in the landscape changed over time? That would be great to show, not just for 2017 (would just need to assess these changes for the 17% of pixels that showed changes and keep the same AGB figures for the remaining 83% of the pixels). This projection would be nice to include in the final version of the manuscript.

Line 270: The Poorter et al. 2016 study is based on trees > 10 cm DBH. This may explain some of the discrepancy. Can you evaluate the contribution of trees 5-10 cm DBH in the total stand ABG? May be useful for comparing results with other datasets from other regions.

---

## Referee Comment (RC2) · Anonymous Referee #2 · 11 Oct 2019

This study combines field, airborne (Lidar) and satellite (Landsat) data to estimate biomass stocks and accumulation in a regenerating tropical forest in Thailand. The authors use multi-temporal Landsat images to identify and target pixels that went from "non forest" to "forest" since 1972. They estimate the rate of biomass gain of these pixels by regressing the recovering time and biomass estimations from a locally calibrated AGB model. Their approach is a clever and effective way to assess biomass dynamics in the absence of multi-temporal biomass maps, and will probably encourage similar future studies in other areas. Recovery from forest disturbance is an important but still poorly understood topic, and this study will definitely contribute to advancing this field. The paper is very well-written, clear and well organized. The methodology is on point and the authors use high quality validation methods, making the whole study very ro-

bust. All the methods used in this study have already been used in various studies, but the way the authors combine them is unique. The paper could be improved by making some minor changes presented below.

General comments:

- My main concern is that only 3 plots with biomass <100Mg/Ha were used to make the AGB lidar model. This is an important point, since the study is focusing on low biomass/recovering areas. This should at the very least be addressed in the Discussion. - Although I understand why the authors chose to focus on pixels that went from "non-forest" to "forest", I think it would be nice to also talk about the pixels that remained "non-forest". There is no information on these in the paper, as the authors are not making any distinction between the "non-changing" pixels (pixels that remained non-forest vs. pixels that remained forest). I think it would be very informative to see how much of the degraded pixels did not recover since 1972 (excluding roads). It would give a more complete picture of the state of the forest. Ignoring these pixels implies that all disturbed areas have recovered. It would be nice to mention this somewhere in the manuscript, and also perhaps making this distinction in Figure 4.

Specific comments:

- L.89: How long before? Is there any historical information indicating when it started? - L.111: "SIS; n=3". This low number should be addressed in the discussion. - L.114: If SES forest is 35-40 years old and OGS forest is more than 200 years old, what is in between? - L.160: "(see below)". Replace by "see Table S1"? - L.199-200: address "pixels that remained non-forest" (see General comment number 1) - L.206-208: This part of the methodology should be explained in more details, or differently. I wasn't sure what you meant until I saw Figure 5. - L.225: separate the pixels that stayed non forest and those that stayed forest - L.228: Do we know why? Is this addressed somewhere? - L.233: Why only eight? Mention Figure S4? - L.270-271: If the rate of accumulation is increasing, shouldn't the rate over 40 years be higher than the one over 20 years?

[Figure]

Figures:

Figure 1: add "(TCH)" in caption Figure 4: add grey classes to legend Suggestion: Make distinction between "pixels that remained forest" and "pixels that remained non-forest"

Supporting information: Table S1: Highlight the results of TCH and mention in caption that it is the best metric. Table S2: Keep same name conventions as in TableS1. Are these the best 4 models? Figure S1: It would be nice to add the sub classes I mentioned about Figure 4, if possible. Figure S2: Compare the histogram presented in Figure 3 to this one.

Also, it would be nice to have some results from the random forest analysis somewhere in the Supporting Information, and add that reference in the main text.

Minor comments:

- L.36: Replace "The previous study" by "A recent study" - L.98: "which the plot officially joined": please rephrase - L.122: Replace "into the ground" by "into ground" - L.138: Replace "cannot" by "could not" - L.211: For consistency, keep the order you present forest classes the same throughout the paper (SIS, SES, OGS)

---

## Referee Comment (RC3) · Rico Fischer (Referee) · 14 Oct 2019

In this study, field data is combined with lidar data to estimate the biomass of secondary forest for a landscape in Thailand. In addition, Landsat scenes were used to distinguish forest and non-forest for the years 1972 to 2017. This made it possible to investigate the biomass dynamic of secondary forest in more detail.

The study is very clearly structured and well written. The methodsTro are applied straightforward. This study is very interesting and very important, even if it was only studied for a small region in Thailand. However, it is not clear to me why the authors did not used existing products and created their own products instead. A new biomass model is calibrated for the region in Thailand (from ALS with TCH as metric) - although

there are already many studies on biomass estimation from ALS available with generalized models. Why calibrating a very specific equation for a small region? The same for the forest/non-forest maps. There are already products available (e.g. from Hansen or Sexton). Why generate your own product? There are certainly good reasons for this, but they should be discussed. In my view, the authors lose the opportunity to generalise this important study and apply them to larger regions.

Further comments:

Title: The title is too general and does not address the specificity of the study – the forest carbon recovery in secondary forests for the last 42 years.

L 160: What are the lidar metrics? "see below" make no sense to me. Please refer to Table S1.

Equation 4: What is the spatial scale of the biomass estimation (AGB\_L)? Is it 0.5ha? Please add the scale and also the r2 beside RMSE.

Fig.2: The abbreviations (SIS, SES, OGS) only become clear if you read the whole text. It would be helpful to briefly explain the abbreviations here as well.

Fig.3: Assuming Eq. 4 gives the biomass at the scale of 0.5ha, how can you generate with this a biomass map with the spatial scale of 60m?

Fig.4: Why is there no transition from forest to non-forest? Could forest loss play a role in the results of the study?

Fig.5: This figure shows that the secondary forest allocates more and more biomass as it becomes older. But if you look at Fig. 3, you can see the highest biomass values between 300 and 400 Mg/ha. Shouldn't Fig.5 therefore show saturation in the AGB recovery at some point? Possibly a power model (red line) is not suitable as a model due to the unlimited growth, but rather a model with a capacity limit.

---

## Author Response (AR1)

Dear Editor,

We appreciated the careful assessment of our manuscript by three highly qualified reviewers and were very pleased by the positive and constructive comments they provided. The reviewers made some useful suggestions to strengthen the manuscript, which we have addressed in the revised manuscript. Below, we provide our responses to the comments raised by reviewers (individually uploaded on the interactive discussion), the revised manuscript and the supplementary information. We highlighted the changes made in response to the reviewers in yellow.

We have also further marked some changes unrelated to the reviewer's comments in green. We now discuss the new IPCC recommendations published by Requena Suarez et al. 2019 during the review process of our manuscript. These new recommendations lead to a decrease by half of the default rate of carbon accumulation in Asian tropical secondary rainforests but was based on very limited data. Because our results do not support these updated rates, we believe that this discussion will attract many readers. Also, some minor mistakes, unrelated to referee's comments, were corrected in red in the revised manuscript.

Finally, the removal of second affiliation of the first author was done to meet the university guidelines towards doctoral publication.

We hope that the corrected manuscript is now suitable for publication in *Biogeosciences*.

Thank you for your time and consideration.

Sincerely yours,

On behalf of the authors,

Nidhi Jha

**Response to the comments by Anonymous Referee #1 on "Forest aboveground biomass stock and resilience in a tropical landscape of Thailand"**

Dear Reviewer,

Thank you for the careful and attentive assessment of our manuscript. We are very pleased with the positive and constructive comments provided for our work. Please find below our point-by-point response to your *italicized* comments.

On behalf of the authors,

**Nidhi Jha**
* * *
*Reviewer #1: This is a robust and well-written study that combines field measurements, multitemporal satellite imagery, and airborne laser scanning data at the landscape scale to estimate rates of biomass accumulation in naturally regenerating forest vegetation in Khao Yai National Park in central Thailand. As such high-quality information is lacking from most regions of Asia, this study will be a landmark case and illustrates how combinations of different data sources can be used to track changes in landscape scale biomass accumulation and carbon storage in the absence of long-term monitoring data from forest sites. I applaud the authors on a job well done.*

Response: Thank you.

*Reviewer #1, C1: One shortcoming of their model is that very few field sites had low ABG values, so the model may not be as accurate at predicting AGB at low levels.*

**Response:** We agree with the reviewer that we do have a limited number of field plots in low-biomass areas. That said, we do not believe that this constitutes a major problem as model uncertainty is expected to be higher for large biomass values than for small biomass values (Zolkos et al. 2013, Remote Sensing of Environment), as suggested by Fig. 2 of our manuscript. Thus, getting a higher representativity of large AGB values in the model is recommended to minimize model calibration errors.

However, we followed the recommendation of Reviewers 1 and 2 and added the following sentence in section 4.1 in revised manuscript.

**"Due to a limited number of field plots in low-biomass areas we were, however, unable to test whether predicting errors vary with AGB or not."**

*Reviewer #1, C2: With the data that they have, the authors estimated the distribution of AGB values across the landscape. These data were used to estimate mean landscape-scale AGB (and carbon density) for 2017. With the information on changing states of pixels from non-forest to forest (or from forest to non-forest), it should be possible to estimate how the distribution, mean,*

*and total AGB within the landscape changed from the mid-1970s to the present day. This would be fascinating to do (if not in this paper, then in another one).*

**Response:** We agree that tracking the AGB distribution over time would be extremely informative and would provide important insights on the carbon balance of the landscape. However, we here face one important limit of our approach that prevent us to assess the landscape scale carbon balance over the study period. Our approach only allows to assess the AGB dynamics of pixels that experienced a single shift from non-forest to forest during the study period. Although this approach generates much more data than usually available through field-based approaches (n=550 in our case), these pixels only represent 4% of the landscape and are thus not representative of the whole landscape carbon dynamics. In an ongoing work, we are adopting another approach where we use field estimates of carbon dynamics and extrapolate them through a LiDAR-based forest successional map to estimate the carbon balance of the landscape. Thus, this objective will be rather achieved in another upcoming paper.

**Additional comments:**

***Reviewer #1, C3:*** *Line 36-38: this statement does not describe what Chazdon et al. 2016 concluded. They found that 40 yr of carbon storage in regenerating forests of lowland regions of Latin American tropics alone offset the past 19 years of carbon emissions from fossil fuel burning and industrial sources from all of Latin America (not total carbon emissions).*

**Response:** We agree with the comment and thank the reviewer for pointing this mistake. We rephrased this statement in the revised manuscript as following:

***"A previous study estimated that 40 years of carbon storage in regenerating tropical forests from Latin America offset the past 19 years of carbon emissionsfrom fossil fuels and industrial production at the scale of Latin America (Chazdon et al., 2016)."***

***Reviewer #1,*** *C4: Line 102: what is the age and prior land use of this secondary forest?*

**Response:** This area is a regenerating successional forest resulting from farming activities (mostly rice cultivation) that stopped in the 1960s. Our analyses suggest that these areas shifted from a non-forest to a forest status in 1975 (see SES4 and SES5 in Fig. S4 of original manuscript, now Fig S5 in revised manuscript).

***Reviewer #1,C5****: Line 112: were there any stands in the understory initiation phase? Some details from Chanthorn et al. 2017 should be included here*

**Response:** None of the plots are in the understory re-initiation stage. To make it clear we added the following information in the revised manuscript after L112 of original manuscript.

***"The classification is based on the framework of Oliver and Larson (1996) who studied successional gradients in temperate forests. Although the original framework considered four***

*successional stages, we did not find any area corresponding to the understory re-initiation stage in the study landscape. Most second-growth forests have regenerated since the Park was established about 40-50 years ago so that older second-growth forests, where understory re-initiation occurs, is very rare in this area. In our study, the SES stage is represented by forest of upto 35-40 years, while other SES area in the landscape may typically range upto 55 years (since 1962), as suggested by some hand drawn historic maps (Smitinand, 1968; Cumberlege & Cumberlege, 1964). On the other hand, OGS forest stands mostly correspond to forests with no obvious sign of human disturbance during the last 100 years (Brockelman, 2011)."*

*Reviewer #1,C6:* Line 155: but only those > 5 cm dbh, right?

**Response:** Yes, correct. For sake of clarity we modified the sentence in the revised manuscript as following:

*"AGB at the plot level was then estimated in Mg ha$^{-1}$ by summing individual tree AGB for all trees with dbh ≥ 5cm belonging to the plot."*

*Reviewer #1,C7:* Line 241: I would take out the word "probably" Why wouldn't it? How has the carbon storage in the landscape changed over time? That would be great to show, not just for 2017 (would just need to assess these changes for the 17% of pixels that showed changes and keep the same AGB figures for the remaining 83% of the pixels). This projection would be nice to include in the final version of the manuscript.

**Response**: Please, see our response to comment C2. Keeping the same AGB values for the 83% pixels would lead to a strong underestimation of the carbon sink in this landscape, as revealed by our ongoing work. We, however, removed the word probably in the Line 241 and add the following sentence in section 3.2 in revised manuscript:

*"Focusing on the 17% pixels that experienced at least one shift from non-forest to forest since 1972, we thus estimate that the study area has stored a minimum AGB of 455 Gg, equivalent to 214 GgC during the study period.".*

*Reviewer #1,C8:* Line 270: The Poorter et al. 2016 study is based on trees > 10 cm DBH. This may explain some of the discrepancy. Can you evaluate the contribution of trees 5-10 cm DBH in the total stand ABG? May be useful for comparing results with other datasets from other regions.

**Response**: Thank you for pointing this issue. According to our field data, the contribution of trees 5-10cm dbh to the total stand AGB ranges from 1% to 39% (average of 4.5%) and tend to decrease with successional stage. Thus, we indeed cannot exclude that part of the difference is due to the inclusion of trees of 5 to 10 cm in dbh. We have added the following sentence in revised manuscript to acknowledge this.

*"After 20 years of recovery, our model predicts an AGB accumulation of 143 Mg ha−1, an estimate slightly higher than the one predicted by Poorter et al., (2016a) in Neotropical secondary forests (122 Mg ha−1). However, this difference can partly be explained by the inclusion of trees between 5 and 10 cm dbh in our study, contrary to Poorter et al. (2016)'s study."*

**Response to the comments by Anonymous Referee #2 on "Forest aboveground biomass stock and resilience in a tropical landscape of Thailand"**

Dear Reviewer,

We are thankful for your careful assessment and dedicated efforts towards improvement of our manuscript. We were very pleased to account for your positive and constructive comments. Please, find below our point-by-point response to your *italicized* comments.

On behalf of the authors,

**Nidhi Jha**

**Reviewer #2:** *This study combines field, airborne (Lidar) and satellite (Landsat) data to estimate biomass stocks and accumulation in a regenerating tropical forest in Thailand. The authors use multi-temporal Landsat images to identify and target pixels that went from "non forest" to "forest" since 1972. They estimate the rate of biomass gain of these pixels by regressing the recovering time and biomass estimations from a locally calibrated AGB model. Their approach is a clever and effective way to assess biomass dynamics in the absence of multi-temporal biomass maps, and will probably encourage similar future studies in other areas. Recovery from forest disturbance is an important but still poorly understood topic, and this study will definitely contribute to advancing this field. The paper is very well-written, clear and well organized. The methodology is on point and the authors use high quality validation methods, making the whole study very robust. All the methods used in this study have already been used in various studies, but the way the authors combine them is unique. The paper could be improved by making some minor changes presented below:*

**Response:** Thank you.

**General comments:**

*Reviewer #2, C1: My main concern is that only 3 plots with biomass <100Mg/Ha were used to make the AGB lidar model. This is an important point, since the study is focusing on low biomass/recovering areas. This should at the very least be addressed in the Discussion.*

**Response C1**: Please see our response to Reviewer #1 comment C1. As argued in our response, we agree that we have a rather limited number of low AGB value field plots, but we do believe that it did not much impact the LiDAR model. As said in the response to reviewer 1, we added the following sentence in section 4.1:

*"Due to a limited number of field plots in low-biomass areas we were, however, unable to test whether predicting errors vary with AGB or not."*

***Reviewer #2, C2:*** *Although I understand why the authors chose to focus on pixels that went from "non-forest" to "forest", I think it would be nice to also talk about the pixels that remained "non-forest". There is no information on these in the paper, as the authors are not making any distinction between the "non-changing" pixels (pixels that remained non-forest vs. pixels that remained forest). I think it would be very informative to see how much of the degraded pixels did not recover since 1972 (excluding roads). It would give a more complete picture of the state of the forest. Ignoring these pixels implies that all disturbed areas have recovered. It would be nice to mention this somewhere in the manuscript, and also perhaps making this distinction in Figure 4.*

**Response C2:** Thank you for this suggestion to which we agree. We thus highlighted areas that remained non-forested since 1972 in Figure 4 of the revised manuscript. About 5% of the study area stayed non-forested since 1972. Most of these areas correspond to areas that were continuously cleared by the Park for administrative or tourism purpose, such as building and camping areas, or wildlife watchpoints. We have added the distinction of the areas as follows which remained forested from all the non-changing pixels in the result section 3.2 in the revised manuscript.

***"Figure 4a illustrates the resulting spatialized time series of non-forest-to-forest shifts over the study area and showed that most (83%) of the landscape did not experience such shift at 60-m resolution, out of which 5% of the area remained permanently non-forested over the 42-year study period. Most of these non-forested areas were continuously cleared and mostly correspond to National Park buildings, including tourist shops and guest houses or camping location"***

***Revised Figure 4***

[Figure]

Specific comments:

***Reviewer #2, C3:*** *L.89: How long before? Is there any historical information indicating when it started?*

**Response C3:** Unfortunately, we do have very limited information on the history of this area. The rough estimate of start of swidden agriculture is about the end of 19th century or early twentieth century (Chanthorn et al., 2016, Theoretical Ecology). For instance, Cumberlege & Cumberlege (1964, VMS Cumberlege Nat. Hist. Bull. Siam Soc 20), who studied orchids in Khao Yai National Park, mentioned that some secondary grassland patches in 1964 were the result of 80 years of swidden agriculture by villagers (hence starting around 1880). All villagers were then expelled by 1962 when the National Park was established (Chanthorn et al., 2017, Forest Ecology and Management). We briefly added estimates in the revised manuscript in section 2.1 as following:

***"Before establishment of the park, some areas were used for low-intensity agriculture activities (Brockelman et al., 2011, 2017) that started at the end of 19th century or early twentieth century (Chanthorn et al., 2016)"***

***Reviewer #2, C4:*** *L.111: "SIS; n=3". This low number should be addressed in the discussion.*

**Response C4:** Please, see our response to Reviewer 1 comment C1.

***Reviewer #2, C5*** *L.114: If SES forest is 35-40 years old and OGS forest is more than 200 years old, what is in between?*

**Response C5:** Please see our response to *Reviewer #1 Comment 5*. Second-growth forests mostly have regenerated since the Park was established about 40-50 years ago. As a consequence, old (50-200 years) second-growth forests are very rare in the landscape so that the understory re-initiation stage is absent from our study. To better explain this, we added the following sentence in the revised manuscript.

***"The classification is based on the framework of Oliver and Larson (1996) who studied successional gradients in temperate forests. Although the original framework considered four successional stages, we did not find any area corresponding to the understory re-initiation stage in the study landscape. Most second-growth forests have regenerated since the Park was established about 40-50 years ago so that older second-growth forests, where understory re-initiation occurs, is very rare in this area. In our study, the SES stage is represented by forest of upto 35-40 years, while other SES area in the landscape may typically range upto 55 years (since 1962), as suggested by some hand drawn historic maps (Smitinand, 1968; Cumberlege & Cumberlege, 1964). On the other hand, OGS forest stands mostly correspond to forests with no obvious sign of human disturbance during the last 100 years (Brockelman, 2011)."***

***Reviewer #2, C6:*** *L.160: "(see below)". Replace by "see Table S1"?*

**Response C6:** We agree, "(see below)" is replaced by "see Table S1" in the revised manuscript.

*Reviewer #2, C7*: *L.199-200: address "pixels that remained non-forest" (see General comment number 1)*

**Response C7:** Please see our response to your C2 comment.

*Reviewer #2, C8:* *L.206-208: This part of the methodology should be explained in more details, or differently. I wasn't sure what you meant until I saw Figure 5.*

**Response C8:** We agree that the sentence was quite unclear and reformulated it as following in the revised manuscript:

**"We thus assigned to each pixel the year of the last non-forest to forest shift, if any, and considered this year as the forest recovery starting time. The AGB predicted by the LiDAR AGB map in 2017 was then used to estimate how much AGB was stored between the forest recovery starting time and 2017 through a non-linear power model."**

*Reviewer #2, C9:* *L.225: separate the pixels that stayed non forest and those that stayed forest.*

**Response C9:** Please see our response to your C2 comment.

*Reviewer #2, C10:* *L.228: Do we know why? Is this addressed somewhere?*

**Response C10:** All cultivated areas were abandoned for natural reforestation after the park establishment in 1962 and a US Army camp was maintained open until the end of the Vietnamese war in 1975 (Chanthorn et al., 2016). By 1990's most of the study areas were thus reforested except few patches such as a golf course that reforested after 2001 (Chanthorn et al., 2016). **Because this historical dynamic is not accounted for in our analyses, we removed this sentence from the revised manuscript**.

*Reviewer #2, C11:* *L.233: Why only eight? Mention Figure S4?*

**Response C11:** We only considered the eight available secondary plots for which forest recovery start felt during the study period. The remaining field plots belong to the old growth forest type and were forested during the whole study period (see also Figure S4 of original manuscript, now Figure S5 in revised manuscript). To be more explicit we slightly reformulate the sentence as followed:

**"Using field AGB estimates at two different census dates from eight secondary forest plots that started regenerating during the study period (see Figure S5) ,we showed that the observed rate of AGB accumulation was similar to the one predicted by our model and also tended to increase with forest age (in blue dots in Fig. 5)"**

***Reviewer #2, C12:*** *L.270-271: If the rate of accumulation is increasing, shouldn't the rate over 40 years be higher than the one over 20 years?*

**Response C12:** Thank you for identifying this counter-intuitive result. To produce those estimates, we used the formula of the model presented in Fig. 5 dividing the AGB predicted after 20 and 40 years by 20 and 40 respectively. If this approach would have been correct with no intercept in the model, you are right that it led to a counter-intuitive result with the existence of an intercept in the model. **Because the model should have an intercept to accurately capture the AGB dynamics, we now only report the rate of C accumulation during the first 20 years of succession to avoid ambiguity**. We revised the line as following in section 4.2:

***"AGB accumulation in our study corresponds to a net carbon uptake of 3.4 Mg C ha−1yr−1 for the first 20 years."***

***Reviewer #2, C13:*** *Figure 1: add "(TCH)" in caption Figure 4: add grey classes to legend. Suggestion: Make distinction between "pixels that remained forest" and "pixels that remained nonforest"*

**Response C13:** Done, thank you.

Supporting information:

***Reviewer #2, C14:*** *Table S1: Highlight the results of TCH and mention in caption that it is the best metric.*

**Response C14:** Done.

***Reviewer #2, C15:*** *Table S2: Keep same name conventions as in TableS1. Are these the best 4 models?*

**Response C15:** We changed the names in Table S2 with the same name as given in Table S1. We tested TCH with additional metrics but adding a second predictor did not reduce the relative LOOCV-RMSE by more than 1% (mentioned in L171 in the original version of manuscript), so only TCH was selected as final predictor to avoid overfitting issues.

***Reviewer #2, C16:*** *Figure S1: It would be nice to add the sub classes I mentioned about Figure 4, if possible.*

**Response C16:** Subclasses from Fig. 4 derived from the time series illustrated in Fig. S1 (now Fig. S2 in revised manuscript). At a given year, the only information that we can report for a pixel is its forest or non-forest status so that we cannot report the sub classes in the new Fig. S2.

*Reviewer #2, C17: Figure S2: Compare the histogram presented in Figure 3 to this one. Also, it would be nice to have some results from the random forest analysis somewhere in the Supporting Information and add that reference in the main text.*

**Response C17:** As recommended, we modified said Figure S2 (now Figure S3 in revised manuscript) by superimposing the density distribution from Figure 3. We also added the results of Random Forest showing the average variable importance in each Landsat sensor in Figure S1 in the revised manuscript. Both the revised Figure and new added Figure S1 is given:

*Revised Figure S2 (Now Figure S3 in revised manuscript)*

[Figure]

*Addition of Figure S1: Random Forest analysis result*

[Figure]

Minor comments:

***Reviewer #2, C18:*** *L.36: Replace "The previous study" by "A recent study"*

**Response C18:** Done.

***Reviewer #2, C19:*** *L.98: "which the plot officially joined": please rephrase*

**Response C19:** We rephrased the sentence as followed:

***"Center for Tropical Forest Science (CTFS) network with which the plot is officially associated since 2009"***

***Reviewer #2, C20:*** *L.122: Replace "into the ground" by "into ground"*

**Response C20:** Done.

***Reviewer #2, C21:*** *L.138: Replace "cannot" by "could not"*

**Response C21:** Done

***Reviewer #2, C22:*** *L.211: For consistency, keep the order you present forest classes the same throughout the paper (SIS, SES, OGS)*

**Reviewer #2, C22:** We have now maintained the consistency of the forest classes as SIS, SES and OGS throughout the revised manuscript.

**Response to the comments by Dr. Rico Fischer on "Forest aboveground biomass stock and resilience in a tropical landscape of Thailand"**

Dear Dr. Fischer,

Thank you very much for the attentive assessment of our manuscript. We really appreciated your positive and constructive comments. Please find below our point-by-point responses to your *italicized* comments.

On behalf of the authors,

**Nidhi Jha**
* * *
*Rico Fischer (RF): In this study, field data is combined with lidar data to estimate the biomass of secondary forest for a landscape in Thailand. In addition, Landsat scenes were used to distinguish forest and non-forest for the years 1972 to 2017. This made it possible to investigate the biomass dynamic of secondary forest in more detail. The study is very clearly structured and well written. The methodsTro are applied straightforward. This study is very interesting and very important, even if it was only studied for a small region in Thailand.*

Response: Thank you

*RF C1: However, it is not clear to me why the authors did not used existing products and created their own products instead. A new biomass model is calibrated for the region in Thailand (from ALS with TCH as metric) – although there are already many studies on biomass estimation from ALS available with generalized models. Why calibrating a very specific equation for a small region? The same for the forest/non-forest maps. There are already products available (e.g. from Hansen or Sexton). Why generate your own product? There are certainly good reasons for this, but they should be discussed. In my view, the authors lose the opportunity to generalise this important study and apply them to larger regions.*

**Response C1:** RF is right that a few generalized LiDAR models were proposed in the literature (reviewed in Réjou-Méchain et al. 2019 Surveys in Goephysics). The most well-known generalized LiDAR model for tropical forests is from Asner et al. (2012, Oecologia), then updated in Asner & Mascaro (2014, Remote Sensing of Environment). If those generalized models may be useful with a limited availability of field data, they convey large systematic errors when transposed to new areas. For instance, they were reported to underestimate AGB by 7% (Jucker et al. 2017 *arXiv preprint*) and 16% (Réjou-Méchain et al. 2015 Remote Sensing of Environment) in two independent sites compared with locally adjusted models. When extensive field data is available locally, as in our case, there is no doubt that a locally-adjusted model is to be preferred, as recommended in Asner et al. (2012) and Asner & Mascaro (2014)'s papers.

A similar reasoning may be applied for the forest/non-forest maps products. While global products such as those proposed by Hansen and Sexton may be useful for some specific, large scale applications, they cannot outperform a locally-calibrated model that was trained with airborne LiDAR data. Even over large spatial scales, Landsat-based Global Forest Watch maps (Hansen)

convey large systematic errors, e.g. a 24% underestimation of gross deforestation at the pantropical scale, with important continental variations (up to 92% in Humid tropical Africa, Tyukavina et al. 2015, Environmental Research Letters). Moreover, most global forest cover maps cover rather limited time periods, e.g. 2000- present for Hansen and 1990-present for Sexton, while we here consider a period of 42 years starting from 1975.

*Further comments:*

**RF C2:** *Title: The title is too general and does not address the specificity of the study – the forest carbon recovery in secondary forests for the last 42 years.*

**Response C2:** We still think that the title well reflects our study as we do also report forest carbon stock estimates. We let the editor decide if we have to change the title.

**RF C3:** *L 160: What are the lidar metrics? "see below" make no sense to me. Please refer to Table S1.*

**Response C3:** As suggested we have replaced "see below" by "see Table S1" in revised manuscript.

**RF C4:** *Equation 4: What is the spatial scale of the biomass estimation (AGB_L)? Is it 0.5ha? Please add the scale and also the r2 beside RMSE.*

**Response C4:** As suggested we have now added the spatial scale (0.5-ha) and the $R^2$ value (0.85) in section 3.1 of the revised manuscript as following:

***"Among all the LiDAR metrics, the mean of top-of-canopy height (TCH, defined as the maximum height of 1-m resolution pixels) was the best predictor of field AGB estimates with a relative RMSE of 14% (RMSE = 45 Mg ha$^{-1}$; $R^2$= 0.85) at 0.5-ha scale"***

**RF C5:** *Fig.2: The abbreviations (SIS, SES, OGS) only become clear if you read the whole text. It would be helpful to briefly explain the abbreviations here as well.*

**Response C5:** We agree. The full form of SIS, SES and OGS abbreviations are now provided in the caption of Figure 2.

**RF C6:** *Fig.3: Assuming Eq. 4 gives the biomass at the scale of 0.5ha, how can you generate with this a biomass map with the spatial scale of 60m?*

**Response C6:** We simply predicted AGB from a TCH estimated at 60-m resolution instead of 70-m resolution. The reason is that we had to calibrate the model with 0.5-ha field data and to produce a map matching the landsat resolution at 60-m resolution. We agree that this is not an ideal situation but, given that resolution remains very close and given that TCH is a mean, we do not believe that this                                  impacted                                  our                                  results.

*RF C7: Fig.4: Why is there no transition from forest to non-forest? Could forest loss play a role in the results of the study?*

**Response C7:** There were only 70 pixels (0.5%) which shows single transition from forest to non-forest (Figure below). These pixels represent areas close to human-impacted areas (e.g., roads and national park tourism areas).

Since our study area is a protected zone, forest loss is very limited and hence does not play an important role in the overall dynamics.

[Figure]

Figure: Pixels in red shows the pixels that underwent Forest to Non-forest shifts (F to NF) with the other selected pixels that has underwent Non-Forest to Forest (NF to F)

***RF C8:*** *Fig.5: This figure shows that the secondary forest allocates more and more biomass as it becomes older. But if you look at Fig. 3, you can see the highest biomass values between 300 and 400 Mg/ha. Shouldn't Fig.5 therefore show saturation in the AGB recovery at some point? Possibly a power model (red line) is not suitable as a model due to the unlimited growth, but rather a model with a capacity limit*

**Response C8:** As discussed in the manuscript, we agree that a saturation should rapidly occur, typically after 50 years. This is the reason why we assume, as previous authors did, that the overall functional form should rather be a sigmoid form. However, our time period stopped at 42 years so that a power model was more adapted to our data. We however agree that this model cannot be used outside the calibration model domain, i.e. for forests older than 42 years.

[revised manuscript text omitted]

Performance comparisons of several LiDAR-derived metrics to infer AGB at 0.5-ha resolution. Metrics (1-17) were calculated directly from the LiDAR cloud dataset and metrics (18-21) were derived from the canopy height model (CHM), which itself derived from the LiDAR cloud data. LOOCV-RMSE is the back-transformed error of the LiDAR-AGB log-log model obtained through a leave-one-out scheme (see methods). The relative RMSE is the ratio of this LOOCV-RMSE to the mean of field AGB. ==From all the metric the mean top-of-canopy-height (TCH) derived from CHM was the best metric selected, highlighted row in table.==

| S.No. | LiDAR metric | LOOCV-RMSE | Relative RMSE (in %) |
|---|---|---|---|
| 1 | $H_{10}$ ($10^{th}$ Percentile) | 93.53 | 29.70 |
| 2 | $H_{25}$ ($25^{th}$ Percentile) | 72.13 | 22.90 |
| 3 | $H_{50}$ ($50^{th}$ Percentile) | 48.73 | 15.47 |
| 4 | $H_{75}$ (75th Percentile) | 50.08 | 15.90 |
| 5 | $H_{95}$ (95th percentile) | 67.78 | 21.52 |
| 6 | $H_{IQR}$ ($HIQR = Q75 - Q25$) | 81.02 | 25.72 |
| 7 | $H_{mean}$ | 47.16 | 14.97 |
| 8 | $H_{sqmean}$ (quadratic mean) | 48.44 | 15.38 |
| 9 | $H_{cv}$ coefficient of variation of all height | 94.79 | 30.10 |
| 10 | Bin95 (Percent of points within Q95) | 93.95 | 29.83 |
| 11 | Bin75 (Percent of points within Q75) | 96.51 | 30.64 |
| 12 | Bin50 (Percent of points within Q50) | 95.54 | 30.33 |
| 13 | Bin25 (Percent of points within Q25) | 95.51 | 30.32 |
| 14 | Hperc10 Percentage of height ranges in 0–10m | 91.76 | 29.13 |
| 15 | Hperc20 Percentage of height ranges in 0–20m | 74.45 | 23.64 |
| 16 | Hperc30 Percentage of height ranges in 0–30m | 74.98 | 23.81 |
| 17 | Hperc40 Percentage of height ranges in 0–40m | 89.75 | 28.50 |
| 18 | TCH (Mean of top of Canopy Height) | 45.2 | 14.35 |
| 19 | CHM_H50 | 47.8 | 15.18 |
| 20 | $CHMH_{relief}$ ((( mean - min) / (max – min)) | 90.12 | 28.61 |

| 21 | CHMSqMean | 46.83 | 14.87 |

**Table S2.** Results from the model selection approach using TCH and any other of the additional LiDAR-based metrics described in Table S1 in a log-log linear model of the form $log\,(AGB) = a + b \times log\,(TCH) + c \times log\,(X)$, where X is the additional metric tested given in the table. LOOCV-RMSE is the back-transformed error of this model obtained through a leave-one-out scheme (see methods). The relative RMSE is the ratio of the LOOCV-RMSE to the mean of field AGB. Adding a second predictor did not reduce the relative LOOCV-RMSE by more than 1, so only TCH was selected as final predictor.

| Log- Log Model | LOOCV-RMSE RMSE | Relative RMSE Relative to mean AGB |
|---|---|---|
| **AGB~TCH** | 45.2 | 14.35% |
| AGB~ **TCH** + Bin 95 | 44.90 | 14.26% |
| AGB~ **TCH** + Bin 95+H10 | 43.86 | 13.96% |
| AGB~ **TCH** + Bin 95+H10+Hperc40 | 45.11 | 14.32% |

**Table S3: Landsat Time-series data used for the study with corresponding validation score**

| S.No | Landsat Mission | Sensor | Date of collection | Validation Score |
|------|-----------------|--------|--------------------|------------------|
| 1 | Landsat 1-3 | MSS | 19/12/1972 | 94.12 |
| 2 | Landsat 1-3 | MSS | 6/1/1973 | 90.69 |
| 3 | Landsat 1-3 | MSS | 13/12/1975 | 92.65 |
| 4 | Landsat 1-3 | MSS | 18/01/1976 | 94.12 |
| 5 | Landsat 1-3 | MSS | 18/11/1978 | 94.61 |
| 6 | Landsat 1-3 | MSS | 1/12/1979 | 96.57 |
| 7 | Landsat 1-3 | MSS | 13/01/1982 | 95.1 |
| 8 | Landsat 4-5 | TM | 9/12/1987 | 94.61 |
| 9 | Landsat 4-5 | TM | 11/12/1988 | 96.57 |
| 10 | Landsat 4-5 | TM | 13/02/1989 | 96.08 |
| 11 | Landsat 4-5 | TM | 5/4/1990 | 98.53 |
| 12 | Landsat 4-5 | TM | 2/11/1991 | 97.06 |
| 13 | Landsat 4-5 | TM | 18/03/1992 | 95.1 |
| 14 | Landsat 4-5 | TM | 23/11/1993 | 96.08 |
| 15 | Landsat 4-5 | TM | 28/12/1994 | 94.12 |
| 16 | Landsat 4-5 | TM | 20/03/1996 | 96.08 |
| 17 | Landsat 4-5 | TM | 20/12/1997 | 91.67 |
| 18 | Landsat 4-5 | TM | 23/12/1998 | 92.65 |
| 19 | Landsat 4-5 | TM | 26/12/1999 | 96.08 |
| 20 | Landsat 4-5 | TM | 12/12/2000 | 95.1 |
| 21 | Landsat 4-5 | TM | 2/3/2001 | 94.61 |
| 22 | Landsat 4-5 | TM | 24/01/2002 | 97.06 |
| 23 | Landsat 4-5 | TM | 21/11/2004 | 97.55 |
| 24 | Landsat 4-5 | TM | 13/03/2005 | 98.04 |
| 25 | Landsat 4-5 | TM | 13/12/2006 | 94.61 |
| 26 | Landsat 4-5 | TM | 30/01/2007 | 95.1 |
| 27 | Landsat 4-5 | TM | 18/12/2008 | 94.12 |
| 28 | Landsat 4-5 | TM | 19/11/2009 | 92.65 |
| 29 | Landsat 4-5 | TM | 25/01/2011 | 95.59 |
| 30 | Landsat 8 | OLI & TIRS | 30/11/2013 | 89.71 |
| 31 | Landsat 8 | OLI & TIRS | 19/12/2014 | 93.14 |
| 32 | Landsat 8 | OLI & TIRS | 2/4/2015 | 91.67 |
| 33 | Landsat 8 | OLI & TIRS | 11/3/2016 | 95.1 |
| 34 | Landsat 8 | OLI & TIRS | 25/01/2017 | 96.57 |

**Figures**

[Figure]

**Fig S1: Random Forest results showing the average variable importance in each Landsat sensors used for classification (a) Average variable importance for Landsat 1-3 (MSS) sensor images (1972–1983) (b) Average variable importance for Landsat 4-5 (TM) sensor images (1984–2011) (c) Average variable importance for Landsat 8 (OLI & TIRS) sensor images (2013-2017)**

[Figure]

**Fig S2: Non-Forest and Forest status across period (1972-2017)**

[Figure]

**Fig S3:** AGB recovery of the pixels that experienced a single shift from Non-Forest to Forest. (a)- Map showing spatialized single shifts from non-forests to forests with the corresponding AGB gain in 2017 as predicted by our LiDAR AGB map (Fig. 3a). The shade gradient represents pixels that did not experience any shift (permanently forested or permanently deforested) and pixels that experienced a shift but that did not pass our quality procedure during the study period (Not selected) (b)- Density distribution of pixels with AGB gain which experiences single shifts over the landscape during the study period compared with the density distribution of predicted AGB over full landscape 2017 (Fig. 3b)

[Figure]

**Fig S43: Distribution of the power coefficients obtained from site-specific power models fitted on AGB recovery versus forest age in 21 sites studied by Poorter et al. (2016) and in our site (red line). We only considered the sites having a minimum of 10 observations and that were younger than 45 years old. We excluded 7 sites matching those rules as they exhibited dubious patterns of carbon recovery through time that cannot be captured by a power model (sites Eastern Pará 2, El Carite, Mata Seca, Patos, San Carlos, Yucatán, Zona Norte).**

[Figure]

**Fig S5.** Non-forest (red) to forest (green) status during the 1972-2017 period in 10 field plots belonging to different successional stages as estimated from our forest classification approach. We did not represent here the subplots belonging to the Mo Singto plot as they all were in a forested status during the whole study period.

---

## Author Response (AR2)

Dear Editor,

Thank you for your feedback and for the careful assessment of our revised manuscript. Please find attached the new revised version of our manuscript that account for your comments. All changes relative to the last revised version are highlighted in **blue**. We apologize for the English mistakes that occurred during the first revision. They have now been corrected, but not highlighted when minor.

We hope that this new version is now suitable for publication in *Biogeosciences*.

Thank you for your time and consideration.

Sincerely yours,

On behalf of the authors,

Nidhi Jha

**Forest aboveground biomass stock and resilience in a tropical landscape of Thailand**

Nidhi Jha1, Nitin Kumar Tripathi1, Wirong Chanthorn2, Warren Brockelman3, Anuttara Nathalang 3,4, Raphaël Pélissier5, Siriruk Pimmasarn1, Pierre Ploton5, Nophea Sasaki6, Salvatore G.P. Virdis1, Maxime Réjou-Méchain5

[revised manuscript text omitted]